# Natural Gallic Acid and Methyl Gallate Induces Apoptosis in Hela Cells through Regulation of Intrinsic and Extrinsic Protein Expression

**DOI:** 10.3390/ijms24108495

**Published:** 2023-05-09

**Authors:** Hasmah Abdullah, Ilyana Ismail, Rapeah Suppian, Nor Munirah Zakaria

**Affiliations:** 1Faculty of Resilience, Rabdan Academy, Al Dhafeer Street, Abu Dhabi 22401, United Arab Emirates; 2School of Health Sciences, Health Campus, Universiti Sains Malaysia, Kubang Kerian 16150, Kelantan, Malaysia; 3Faculty of Health Sciences, Universiti Sultan Zainal Abidin, Gong Badak Campus, Kuala Nerus 21300, Terengganu, Malaysia

**Keywords:** apoptosis, gallic acids, methyl gallate, intrinsic and extrinsic pathways

## Abstract

Induction of apoptosis is one of the targeted approaches in cancer therapies. As previously reported, natural products can induce apoptosis in in vitro cancer treatments. However, the underlying mechanisms of cancer cell death are poorly understood. The present study aimed to elucidate cell death mechanisms of gallic acid (GA) and methyl gallate (MG) from *Quercus infectoria* toward human cervical cancer cell lines (HeLa). The antiproliferative activity of GA and MG was characterised by an inhibitory concentration using 50% cell populations (IC_50_) by an MTT [3-(4,5-dimethylthiazol-2-yl)-2,5-diphenyl tetrazolium bromide] assay. Cervical cancer cells, HeLa, were treated with GA and MG for 72 h and calculated for IC_50_ values. The IC_50_ concentration of both compounds was used to elucidate the apoptotic mechanism using acridine orange/propidium iodide (AO/PI) staining, cell cycle analysis, the Annexin-V FITC dual staining assay, apoptotic proteins expressions (p53, Bax and Bcl-2) and caspase activation analysis. GA and MG inhibited the growth of HeLa cells with an IC_50_ value of 10.00 ± 0.67 µg/mL and 11.00 ± 0.58 µg/mL, respectively. AO/PI staining revealed incremental apoptotic cells. Cell cycle analysis revealed an accumulation of cells at the sub-G1 phase. The Annexin-V FITC assay showed that cell populations shifted from the viable to apoptotic quadrant. Moreover, p53 and Bax were upregulated, whereas Bcl-2 was markedly downregulated. Activation of caspase 8 and 9 showed an ultimate apoptotic event in HeLa cells treated with GA and MG. In conclusion, GA and MG significantly inhibited HeLa cell growth through apoptosis induction by the activation of the cell death mechanism via extrinsic and extrinsic pathways.

## 1. Introduction

Cervical cancer ranks as the fourth most frequently diagnosed cancer incidence for women globally. In 2018, an estimated 570,000 women were diagnosed with cervical cancer worldwide, and about 311,000 women died from the disease [1]. Effective primary (Human papillomavirus (HPV) vaccination) and secondary prevention approaches (screening for and treating precancerous lesions) will prevent most cervical cancer cases. It begins in the cervix and will invade other tissues near the cervix and organs such as the lungs or liver. Signs and symptoms of cervical cancer are irregular menstruation, weight loss, heavy menstruation, abnormal menstruation, vaginal discomfort and pelvic pain [2]. Human papillomavirus (HPV) is known as the major risk factor for cervical cancer. Chronic viral infection with high-risk HPV genotypes is the causative agent, which can be observed in 99.7% of cervical cancer patients worldwide [3].

Over the years, researchers and clinicians have approved and established numerous treatments in treating various cancer diseases. Different treatments are given to the patients based on different stages and diagnoses of cancer [4]. However, the use of a single therapy to treat these cancers is less effective because cancer is a complex disease that involves many processes and mechanisms related to the uncontrollable development of the cells [5,6]. There are a few existing treatments for cervical cancer, such as surgery, chemotherapy, radiation and chemoradiation.

Although these therapies have improved the long-term survival of cancer patients, they can cause serious and long-lasting side effects, such as the development of secondary cancers and infertility [7]. Therefore, an increase in resistance to, as well as adverse effects of, cancer treatments has led researchers to investigate novel cancer chemoprevention from herbal resources, which could be used for the effective treatment of cancer diseases [8]. Targeted cancer treatment also became one of the approaches for the effective usage of chemotherapeutic drugs in cancer treatments. Success therapy in cancer treatment is determined by several modes of cell death, such as apoptosis, autophagy and necrosis. Those mechanisms involved the removal of cancer cells [9]. Apoptosis is also called programmed cell death, which is the fastest form of cell death. Apoptosis loss is commonly seen in most drug-resistant cancers. Hence, the induction of apoptosis in the target cancer cells is the therapeutic goal of any cancer therapy [10].

Under critical physiological conditions, different endogenous tissue-specific agents and exogenous cell-damaging agents initiate programmed cell death in a particular cell type [11]. The mechanism of action of apoptotic cell death is typically characterised by chromatin condensation, DNA fragmentation, cell shrinkage, membrane blebbing and loss of adhesion to extracellular matrices. The externalisation of phosphatidylserine, and the activation of cysteine aspartyl proteases, called caspases, feature biochemical changes in cell death [12]. Therefore, recent advances in cancer research are focused on the development of new drugs that halt the escape behaviour of cancer cells via the execution of apoptosis. Failure to induce apoptosis can eventually lead to the proliferation of cancer cells. Moreover, the resistance of apoptosis stimulates aberrant cellular multiplication, which eventually leads to tumourigenesis and is a significant obstacle in cancer treatment [13]. The mechanism of apoptosis involves extrinsic and intrinsic pathways [14]. The extrinsic pathway refers to the death receptor-mediated pathway, and the intrinsic pathway is mitochondrial-mediated [14]. Interestingly, most cytotoxic drugs and various extra- and intra-cellular stresses trigger the intrinsic pathway.

In the continuous search for safer and more effective cancer treatments, plant-derived natural compounds are of major interest [15,16]. In this regard, plant phenolic secondary metabolites are among the candidates, which have drawn much attention in anticancer drug development and many evidence potential as anti-cancer compounds [17]. Phenolics display a great prospective as cytotoxic anti-cancer agents by promoting apoptosis, reducing proliferation, and targeting various aspects of cancer (angiogenesis, growth and differentiation, and metastasis) [18,19].

Gallic acid and its methyl ester, methyl gallate, are two common phenolic acid compounds found in various plants and herbs. Both are well known to have strong antioxidant properties, as their antioxidant activity has been reported in numerous studies [19]. To date, there are several biological activities of gallic acid and methyl gallate described, including antimicrobial activity, antifungal [19], anti-inflammatory, cardioprotective, gastroprotective and neuroprotective effects [20], and antitumour properties. Both have demonstrated potent antiproliferative activity toward different cancer cell lines, including HeLa cells. Gallic acid and Methyl gallate, in combination with cisplatin, were reported to reduce the proliferation of cervical cancer cells, HeLa, at lower doses [21].

Gallic acid (3,4,5-trihydroxybenzoic acid; GA) is a common phenolic acid compound found in various plants and herbs, such as tea, grapes, gall-nuts and red wine [22,23,24]. It is the most commonly known hydrolysable tannin or gallotannin. In addition to its biological activity, it has the capability to precipitate proteins and form complexes with toxic metal ions, which can reduce their bioavailability in the environment [24]. In the anticancer research on gallic acid, more focus has been given to antitumour capacity on different types of cancer cell lines, including oral, lung, pancreatic and cervical cancer cells [24,25]. Most of the studies reported the ability of gallic acid to regulate apoptosis as an indicator of antitumour effects.

Gallic acid methyl ester, methyl gallate (MG), methyl-3,4,5-trihydroxybenzoic acid, is another phenolic compound abundantly found in plants [26]. A broad range of bioactivity, such as anti-spasmodic, anti-atherogenic, anti-inflammatory and anti-microbial activities, has been reported for this compound [27,28,29]. Previously, methyl gallate was reported to decrease oxidative stress and DNA damage related to hydrogen peroxide in MDCK cells [27]. It reduced lipid peroxidation and prevented the depletion of intracellular glutathione (GSH) [27]. In the HaCaT normal skin cell line, methyl gallate showed low cytotoxic effects [30], indicating its safety in general.

The anticancer properties of natural products are generally linked to their antioxidant capacity and mode of cell death, apoptosis. In this regard, the antioxidant properties of gallic acid and its derivative, methyl gallate, together with their chemoprevention properties toward various types of cancers, are strongly supported [31]. Studies on different types of cancer cell lines with GA and MG mostly focused on anticancer activity, with limited knowledge provided on the mechanism of apoptosis. The understanding of how GA and MG induce cell apoptosis is still limited. Taking these findings into consideration, the present study hypothesised and investigated whether two natural phenolic acids, MG and GA (isolated from *Quercus infetoria* galls extract), exert anticancer effects on HeLa cells by modulating the expression of extrinsic and intrinsic factors involving downstream apoptotic molecules, including B-cell lymphoma 2 (Bcl-2) and Bcl-2-associated X protein (Bax) and the extrinsic caspase 8 and 9.

## 2. Results

### 2.1. Bioassay Guided Isolation of EAQI

The EAQI extract exerted a potent antiproliferative effect toward HeLa cells with an IC_50_ value of 11.50 ± 0.50 µg/mL after 72 h of treatment (Table 1), and no effect was observed toward normal cells, Vero.

The EAQI crude extract was further purified to isolate active compounds by means of bioassay-guided isolation. Four main fractions from the first round of the column chromatography (CC) were eluted and named EAQI/FA1, EAQI/FA2, EAQI/FA3 and EAQI/FA4. An antiproliferative assay for each fraction was conducted using an MTT assay on Hela and Vero cells. From the results, EAQI/FA3 was the most active fraction with IC_50_ values of 16.67 ± 1.76 µg/mL. The fraction EAQI/FA3 was subsequently subjected to the second CC for further purification, resulting in five more subfractions (Table 2). Two pure compounds were isolated from subfractions EAQI/FA3/B2 and EAQI/FA3/B5. Both compounds demonstrated potent antiproliferative activity towards HeLa cells with an IC_50_ value of 11.00 ± 0.58 µg/mL and 10.00 ± 1.06 µg/mL, respectively (Table 2). Interestingly, active fractions and subfractions exerted a cyto-selective effect on HeLa cancer cells. Both subfractions EAQI/FA3/B2 and EAQI/FA3/B5 were further resumed for structural elucidation and analysis to identify the compounds.

### 2.2. Structure Elucidation

#### 2.2.1. EAQI/FA3/B2 (Methylgallate, MG); Methyl 3,4,5-Trihydroxybenzoate

The subfraction EAQI/FA3/B2 was isolated as a pure white solid compound. The molecule structure of EAQI/FA3/B2 was analysed by 1D and 2D NMR spectroscopy. The 1H NMR spectrum revealed a doublet signal at δH 6.95 ppm corresponding to two protons, H2 and H6, which were symmetrical. Hence, both protons exhibited the same signal at δH 6.95 ppm. At the higher field region, there was another signal at δH 3.73 ppm corresponding to an aliphatic methoxy proton. Based on the signal integration ratio, the presence of these protons was attributed to the number of protons in the methoxy group.

The 13C NMR spectrum exhibited the presence of eight carbon peaks at δC 169.05, 146.55 (representing two carbons), 139.79, 121.46, 110.06 (two carbons) and 52.29 ppm. The carbon peaks consisted of one methyl carbon, three quaternary carbons, an oxygenated aromatic carbon signal at δC 139.79 ppm and one carbonyl carbon at δC 169.05 ppm. Analysis of HMBC and HMQC spectra demonstrated the correlation between carbonyl ester and the aromatic proton signal.

Infrared analysis showed that the adsorption wavelength at 1691.41 cm^−1^ indicated the presence of carbonyl group stretching (C=O). Meanwhile, 2960.66 cm^−1^ demonstrated the C-H stretching (alkane) of the methyl group. In addition, the adsorption wavelength at 3515.62 cm^−1^ and a broad peak at 3373.44 cm^−1^ corresponded to the presence of a hydroxyl group in the compound (O-H). Then, the strong absorption wavelength at 1618.52 cm^−1^ revealed the presence of C=C stretching in the benzene ring.

The mass spectrum for this compound (EAQI/FA3/B2) exhibited a molecular ion peak of *m*/*z* 184, which corresponded to the molecular weight formula of C*_8_*H*_8_*O*_5_*. Fragment ions were observed at *m*/*z* 125 due to the loss of -CO from the ion *m*/*z* 153. Meanwhile, *m*/*z* 153 was exhibited due to the loss of –OCH*_3_* from *m*/*z* 184 [M+]. This spectrum demonstrated the presence of the ester group (COOCH*_3_*). In addition, fragment ions were observed at *m*/*z* 107, *m*/*z* 79 and *m*/*z* 51 due to the loss of H*_2_*O and –CO from the benzene ring. This result confirmed the presence of three hydroxyls on the benzene ring. The occurrence of these results identified the EAQI/FA3/B2 compound as methyl gallate (MG). The International Union of Pure and Applied Chemistry (IUPAC) name is methyl 3, 4, 5-trihydroxybenzoate with chemical formula C*_8_*H*_8_*O*_5_*.

#### 2.2.2. EAQI/FA3/B5 (Gallic Acid, GA); 3,4,5-Trihydroxybenzoic Acid

EAQI/FA3/B5 (78.7 mg) was isolated as a white crystal. It showed brown spots with iodine vapour staining and FeCl*_3_* spraying reagent. The dark spot was observed under UV-254 nm. The molecule structure of gallic acid was analysed with 1D and 2D NMR spectroscopy. The spectrum of 1H NMR revealed a single signal at δH 7.078 ppm, corresponding to two protons (H1 and H5). Meanwhile, the 13C NMR spectrum indicated the presence of seven carbon peaks at δC 169.21, 146.40 (2 carbons), 139.84, 121.47 and 110.17 (two carbons) ppm. The carbon peaks consisted of two methine carbons, four quaternary carbons and a carbonyl carbon. A correlation between H1 and H5 with carbon in the aromatic ring (C4, C6, C5, C2, C3 and C7) was identified in the HMBC and HMQC spectra.

The infrared spectrum demonstrated a broad peak at 3285.83 cm^−1^, corresponding to the presence of a carboxylic group in the compound. The frequencies of 3370.12–3064.88 cm^−1^ indicated the presence of a phenolic group, specifically an aromatic group, attached to the hydroxyl group. The strong absorption wavelengths at 1620.84 cm^−1^, 1620.84 cm^−1^ and 1541.40 cm^−1^ demonstrated the presence of a benzene ring system (C=C). The strong absorption at 1702 cm^−1^ was related to carbonyl group stretching (C=O). In addition, the absorption bands at 1248.56 cm^−1^, 1026.93 cm^−1^ and 1026 cm^−1^ indicated the presence of three O-aryl groups that were attached to the benzene ring. Two of the O-aryl groups were identical, known as meta carbonyl groups.

The mass spectrum showed a molecular ion peak of *m*/*z* 170, which corresponded to the molecular ion formula (C_7_H_6_O_5_). The presence of the main ions at *m*/*z* values 153, 135, 125, and 107 was due to the loss of acid and hydroxyl functional groups. Meanwhile, the fragmentation ions at *m*/*z* values 96, 79, 68, 51, and 39 represented the characteristics of aromatic ring breakdown. The occurrence of these results confirmed the presence of the gallic acid compound. The IUPAC name is 3,4,5-trihydroxybenzoic acid with chemical formula C_7_H_6_O_5_.

### 2.3. Antiproliferative Activity Gallic Acid and Methyl Gallate

Gallic acid and methyl gallate both demonstrated high antiproliferative activity toward HeLa cells with an IC_50_ value of 10.00 ± 1.06 µg/mL and 11.00 ± 0.58 µg/mL, respectively (Table 3). No antiproliferative activity was observed in the treated Vero cells. These findings indicated that MG and GA are cyto-selective toward cancer cells without affecting the growth of normal cells, Vero. Cisplatin served as a positive control drug and exhibited cytotoxicity effect not only towards HeLa but also Vero cells at lower concentrations.

### 2.4. Mode of Cell Death in the Treated Cells

As shown in Figure 1, the untreated (UT) cells with an intact nucleus were mostly green in colour, indicating healthy viable cells. In general, cells treated with GA and MG exhibited the same apoptotic characteristics. Early apoptotic features were seen in GA- and MG-treated cells, such as cell blebbing, fragmented DNA and chromatin condensation, after 24 h. Late apoptotic cells, marked with an orange colour, were obviously seen at 24, 48 and 72 h. Necrotic cells (stained red by PI) were mostly observed at 72 h. Most of the small apoptotic bodies were identified at 48 and 72 h. The untreated (UT) cells with an intact nucleus were mostly green colour indicating healthy viable cells. All the treated cells exhibited early apoptosis features with cell blebbing, fragmented DNA and condensed chromatin after 24 h of treatment. The late apoptosis with orange colour were also detected at 24, 48 and 72 h of treatment. However, more necrotic cells (stained red by PI) were most observed at 72 h of treatment with GA, MG and CIS. Small apoptotic bodies were identified at 48 and 72 h of treatment. In all treatment durations, the relative number of green cells decreased then slowly increased with incubation time, suggesting the change from viable to apoptosis.

The Apoptosis event in all treated cells significantly increased in a time-dependent manner (*p* < 0.05) compared to UT (Figure 2). The percentages of apoptotic cells increased from 12.3% (24 h) to 15.3% (48 h) and 21.7% (72 h) in HeLa cells treated with GA. No significant difference in apoptotic cell percentages (*p* > 0.05) was observed between GA and MG treatment. On the other hand, necrotic cells were also observed in the treated group, which might be due to the progression of the apoptotic cells into secondary necrosis.

### 2.5. Detection of Phosphatidylserine (PS) Externalisation

Quantification of annexin V-FITC binding to externalised PS represents the apoptotic cells. In this study, annexin V-FITC was plotted versus HeLa cell distribution within four different quadrants (Figure 3). The lower left quadrant of the fluoro cytogram shows viable cells (Q3), whereas the lower right quadrant shows early apoptotic cells (Q4). The upper right quadrant shows late apoptotic cells (Q2) and the upper left quadrant shows necrotic cells (Q1). Obviously, cell populations in all treated groups tended to shift from viable to early apoptosis as early as 3 h compared to the UT group.

In the UT group, only 0.1% to 0.24% of cells were shifted to a necrosis event (Q1) after 12 h of treatment. This showed that the cells which underwent annexin V-FITC analysis were at high viability and in a healthy condition. In MG-treated cells, the early apoptosis event increased from 2.36% to 2.5% and 2.67% for a treatment duration of 3, 6 and 12 h, respectively. Meanwhile, the early apoptosis event in GA-treated cells increased from 2.24% (3 h) to 3.2% for 6 h and 3.8% for 12 h of treatment. In addition, the percentages of cells in the late apoptosis event (Q2) and necrosis (Q1) for all treated groups also increased in a time-dependent manner. However, the percentage of necrotic cells observed in all treatment groups significantly increased (*p* < 0.05) compared to early and late apoptosis after 12 h of treatment. This condition may be due to the transition of apoptotic cells into secondary necrotic cells after a certain time. The absence of the phagocytic process in cultured cells will lead to secondary necrosis, which is a post-apoptotic event.

On top of that, the percentages of cells in late apoptosis event (Q2) and necrosis (Q1), for all treatment groups were also increased in a time dependent manner (Figure 4). However, the percentage of necrotic cells (Q1) in all treatment groups had increased significantly (*p* < 0.05) compared to early and late apoptosis after 12 h of treatment. This condition may be due to the transition of apoptotic cells into secondary necrotic cells after a certain time period. The absence of phagocytic process in cultured cells will lead to secondary necrosis that is a post-apoptotic event. These findings supported that MG and GA induced cell death by apoptosis pathway.

### 2.6. Cell Cycle Arrest

The results indicated that the subG0 population in all treated cells was significantly increased (*p* < 0.05) compared to UT in a time-dependent manner. In the untreated cells, the cell population arrested in the subG0 phase was only 4.48%, 7.75% and 8.5% after 24, 48 and 72 h, respectively (Figure 5).

GA treatment showed the highest percentage of the subG0 population, which was 7.43% after 24 h. The percentage was higher than MG (6.54%) and CIS (5.76%). Interestingly, a higher percentage of cells (26.13%) was observed in the subG0 after exposure to MG for 48 h. In other treatment groups, GA also exhibited a marked increase in cell populations in the subG0 after 48 h. The subG0 population increased further after 72 h treatment with MG (62.36%), followed by GA (36.08%), EA (32.04%) and CIS (24.05%). However, the treatment with MG, GA and CIS did not affect the cell cycle distribution at the G0G1, S and G2/M phases. The increase in cell count in the subG0/G1 phase indicated that the cells undergo apoptosis and associated DNA cleavage [27]. These findings suggested that MG and GA treatment induce apoptosis, as the subG0 peak is reported to be a quantitative apoptosis marker.

### 2.7. Expression of Apoptotic Protein

In this study, the expression of tumour suppressor proteins, p53, Bax (proapoptotic) and Bcl-2 (antiapoptotic) proteins was evaluated using flow cytometry analysis. The expression of p53 proteins after treatment with MG, GA and CIS are shown in Figure 6A,B. The expression of p53, was in accordance with the shifted cell populations in the subG0 to the right compared to UT group.

Meanwhile, Bax proteins were also detected in all treatment groups (Figure 7A,B), whereas Bcl-2 proteins were not expressed (Figure 8A,B). This finding suggested that MG and GA induced apoptosis by elevating the expression of Bax and p53 and suppressing the expression of Bcl-2.

### 2.8. Caspase Analysis

The cells were exposed to FAM Flica reagent and were sorted into two different cell populations accordingly. The populations with inactive caspases were situated to the left and the active caspases shifted to the right.

In this study, cells treated with MG and GA expressed caspase 8 and caspase 9. In comparison to the UT group, the 3 h treatment of HeLa cells with MG and GA significantly increased (*p* < 0.05) both caspase activity. (Figure 9A,B). The highest percentage of activity for both caspases was seen in cells treated with GA, at 50.20% and 54.26%, respectively, for caspases 8 and 9. For MG, there were not many differences in the activity of the two caspases, 44.08% for caspase 8 and 44.10% for caspase 9. Both caspases were expressed in the MG and GA-treated cells, indicating that both involved extrinsic and intrinsic apoptotic pathways.

## 3. Discussion

The extract of *Quercus infectoria* (QI) was previously reported to contain 50 to 70% tannin, with a small amount of free gallic acid and ellagic acid [32]. Reviews have reported that the main bioactive phytochemicals of QI are phenolic compounds. Phenolic compounds are one of the largest groups of secondary metabolites in plants, with great importance due to their occurrence and pharmacological properties [33,34]. A study by [35] demonstrated that the total amount of phenolic compound was 57.5% in methanol extracted from QI. Therefore, high amounts of phenolic compounds present in the extract of QI implied that they might be the active compound responsible for the treatment and prevention of various diseases [32,36].

The development of cancer is a multi-stage process that involves initiation, promotion and progression. Dietary polyphenols can affect and modulate multiple diverse biochemical processes and pathways involved in carcinogenesis. Several cancer preventive mechanisms have been identified as affected by polyphenols, including prevention of oxidation, detoxification of xenobiotics and induction of apoptosis. Most investigations have been conducted with individual polyphenols in order to increase our understanding of the biological and cellular mechanisms of their anticancer efficacy [33,37].

In the present study, gallic acid (GA) and methyl gallate (MG) exhibited a cyto-selective effect, but did not affect the growth of normal cells, Vero. Previously, Mamat et al. [20] reported the cyto-selective effect of GA and MG toward normal cells, NIH 3T3, at IC_50_ concentrations of both compounds [34,38]. However, the IC_50_ concentration of GA and MG in this study was slightly higher (13.44 and 16.55 µg/mL, respectively), as they used synthetic GA and MG compounds [34,38]. Further study is needed to compare the discrepancies between natural and synthetic compounds.

A study by Chaudhuri et al. [39] demonstrated that MG isolated from *Spondias pinnata* bark also exhibited a potent antiproliferative effect toward human glioblastoma cells, U87 with an IC_50_ value of 8.44 ± 0.61 μg/mL [30,35]. Similarly, GA and MG isolated from the seed coats of *Givotia rottleriformis* reduced cell proliferation with an IC_50_ value of 11 μg/mL and 43 μg/mL at 48 h, respectively, toward human epidermoid carcinoma (A431) skin cancer cells [30]. In the same way, normal skin HaCaT cells treated with GA and MG showed less cytotoxicity than A431 cancer cells with an IC_50_ value of 84.2 μg/mL and 79.4 μg/mL [30].

Consistent with previous reports, the IC_50_ of MG isolated from *Mangifera pajang* exhibited an antiproliferative effect on HeLa cells with an IC_50_ slightly higher than our current findings, which was 18.07 μg/mL [37,40]. In addition, MG isolated from the leaves of the Nigerian medicinal plant, *Anacardium occidentale* L. (Anacardiaceae), also demonstrated higher IC_50_ towards HeLa cells with an IC_50_ value of 49 ug/mL [38,41]. Conversely, a study by Fiuza et al. showed that MG did not inhibit the proliferation of HeLa cells [39,42].

You et al. (2010) reported that GA reduced HeLa cells’ viability with an IC_50_ of 14 μg/mL for 24 h of treatment [40,43]. They also found that GA was selective toward HeLa cells, where a higher IC_50_ of GA was found in human umbilical vein endothelial cells (HUVEC) (IC_50_ = 68 μg/mL). These data were in agreement with the findings from Park (2017), indicating that GA exhibited significantly selective cytotoxicity toward HeLa cells compared to HUVEC [44]. Consistent with previous reports on the antiproliferative effect of GA in HeLa cells, Zhao and Hu (2013) demonstrated that GA isolated from Chinese galls significantly decreased the cell viability of HeLa in a dose-dependent manner [45]. Compared with the effect on the cervical cancer cells, GA showed a substantial reduction in the antiproliferative effects on normal HUVEC at the same concentrations.

The antiproliferative effect of HeLa cells on treatment with GA and MG indicated that apoptosis was a possible mode of cell death. The induction of apoptosis in the targeted cancer cells is the therapeutic goal of any cancer therapy. In this study, AOPI staining and detection of phosphatidylserine (PS) externalisation analysis were used to determine the mode of cell death of HeLa cells. Observations of morphological changes in HeLa cells stained with AOPI revealed clear apoptotic changes and exhibited the common characteristics of apoptotic cell death, such as chromatin and nuclear condensation, cell shrinkage, DNA fragmentations and the formation of apoptotic bodies, compared to untreated cells (UT). The UT group demonstrated healthy viable cells with a green, intact nuclear structure.

Early apoptosis with cell blebbing and nuclear chromatin condensation was observed after 24 h of treatment. Blebbing is a normal cellular activity observed during mitosis. In damaged cells, the presence of blebs illustrates impending cell death and apoptotic cells, which are unable to stop blebbing and flatten back onto the substratum [46]. Furthermore, in the late stages of apoptosis, changes, such as the presence of a reddish-orange colour due to the binding of acridine orange to denatured DNA, were observed and indicated a loss of cell membrane integrity [47].

The apoptotic and necrotic cell populations increased in a time-dependent manner. However, after 72 h, necrotic cells increased significantly (*p* < 0.05) in cells treated with GA and MG compared to UT. This situation may be due to the progression of apoptotic cells into secondary necrotic cells after 72 h. The term secondary necrosis refers to a process in which late-stage apoptotic cells that failed to be engulfed by phagocytes or neighbouring cells undergo necrosis. Secondary necrosis, thus, is a post-apoptotic event. This process is seen in cultured cells undergoing cell death by apoptosis in vitro, induced, for example, by the absence of survival factor signals or activation of death receptors by different lethal signals. In the absence of phagocytic cells that could engulf them, they ultimately cease to be metabolically active, lose membrane integrity and release their cytoplasmic contents into the culture medium [46,48,49].

An apoptotic mode of cell death induced by GA and MG toward HeLa cells was further confirmed by the detection of phosphatidylserine (PS) externalisation using AnnexinV-FITC and PI staining [30]. In this study, HeLa cells treated with GA and MG exhibited a shift in the pattern of the phosphatidylserine (PS) externalisation from viable to early apoptosis and late apoptosis at 3, 6 and 12 h. Similar to AOPI analysis, after 12 h of treatment, the percentage of necrotic cells increased significantly compared to early and late apoptosis. This phenomenon is referred to as secondary necrosis. According to Silva (2010) secondary necrosis is the natural outcome of fully developed apoptosis in multi- and unicellular eukaryotes [50]. Apoptosis was seen to progress until terminal disintegration by secondary necrosis due to the absence of phagocytic cells in vitro. In addition, secondary necrosis might be due to the activation of caspase 3, which triggers necrotic signalling and contributes to inflammation and other immune responses if the dying cells are not quickly removed by phagocytes [51].

Similar findings were reported by Rahim et al. (2018), as they observed that the treatment with MG isolated from Bambangan (*Mangifera pajang*) increased the percentage of apoptotic cells toward MCF-7 cells [40]. Chaudhuri et al. (2015) suggested that MG treatment induced apoptosis but not necrosis in U87 cells [39]. Their results showed an increase in early apoptotic cells with an increase in the dose of MG. At a zero dose of MG, 4.12% of cells were found in the early apoptotic phase, while 4.46%, 5.63%, 28.12%, 43.63% and 50.08% cells were found in the early apoptotic phase with an increase in dose to 10 μM, 25 μM, 50 μM, 75 μM and 100 μM, respectively.

A study by Sourani et al. (2016) demonstrated that treatment with GA also increased the apoptotic cells percentage to 80% in Jurkat cells after 48 h of incubation [52]. Sun et al. (2016) reported that GA induced apoptosis toward SMMC-7721 cells using Annexin-V FITC/PI assays [53]. Similar findings have been reported by Liang et al. (2014) as they observed that Hoechst 33258 and Annexin V/PI staining assays exhibited apoptosis induction by GA toward human chondrosarcoma cells (SW1353) [54]. Untreated cells exhibited a distribution of the stain and round homogeneous nuclei features, whereas apoptosis in treated cells increased gradually in a dose-dependent manner and showed changes typical of apoptosis, including a reduction in cellular volume, bright staining and condensed or fragmented nuclei.

Meanwhile, the Annexin V/PI double-positive populations indicated cells undergoing early or late apoptosis. The percentage of cells undergoing apoptosis (including the early and late apoptosis) with GA treatment was significantly higher than that in untreated cells. Liu et al. (2012) tested the effect of GA on human pancreatic cancer cell lines CFPAC-1 and MiaPaCa-2, as well as hepatocytes HL-7702 normal cells [55]. Typical morphological changes of apoptosis, including chromatin condensation and nuclear fragmentation, were observed in the treated cells compared to normal cells. They suggested that GA induced apoptosis in human pancreatic cancer cell lines.

Flow cytometry analysis of this study showed that MG and GA induced subG0 arrest in a time-dependent manner. As compared to UT group, the percentage of cells in the subG0 peak increased significantly (*p* < 0.05) in all treated groups. Hence, this increased accumulation of cells was much more evident in MG-treated cells. This was followed by a concurrent reduction in all treated groups of the cells in the G0G1 phase due to the inhibition of cell growth that causes cells to enter the subG0 phase [46,56].

Notably, the increase in the subG0 population in this study revealed the induction of apoptosis, as the sub G0 peak is reported to be a quantitative indicator of apoptosis as well as a characteristic hallmark of apoptotic cells [57,58,59]. In addition, the increase in the subG0 population indicated that the nucleus of DNA had been cleaved into multiple fragments, demonstrating induced apoptosis cell death [46]. DNA fragmentation-related cell death produces apoptotic cells with less DNA than healthy cells, resulting in a subG0 peak in a cell population profile [60,61,62,63].

According to Surova and Zhivotovsky (2013), mild DNA damage typically contributes to cell cycle arrest, while severe and irreversible injury changes the cellular response to apoptosis-inducing cell death [64]. Thus, the apoptosis of HeLa cells treated with MG and GA in the current study was confirmed by the cell cycle profile, demonstrating the increase in subG0 populations. These data are in line with earlier published data showing that MG isolated from *Spondias pinnata* demonstrated an increased accumulation of the subG0 population that refers to the apoptotic cell death tested in human glioblastoma cells, U87. The apoptosis was found to increase dose-dependently with nearly 40% of cells in subG0 at a concentration of 100 μM [39]. Similarly, a study by Faried et al. (2007) demonstrated that GA isolated from *Phaleria macrocarpa* increased the number of subG0 populations after 12 h of treatment (14.8%) and increased the peak level to 98.8% after 24 h [65]. Their result revealed that GA increased the number of apoptotic cells in a time-dependent manner.

Contrary to this finding, MG isolated from Bambangan (*Mangifera pajang*) in MCF-7 cells demonstrated cell cycle arrest at G0/G1 and increased cell populations in the subG0 phase indicating apoptosis [40]. Kim et al. (2016) also reported that MG had no effect on the cell cycle regulation of malignant mouse cells, EL4, after 24, 48 and 72 h of treatment [66]. Contrary to this, GA also has been reported to induce G2/M arrest in HeLa cells [9] and human non-small-cell lung cancer NCI-H460 cells [67].

Meanwhile, GA was found to induce G0G1 arrest in human leukaemia cells, HL-60 [68]. The difference in cell cycle arrest regulation of MG and GA seen in HeLa cells and other cancer cell lines was mainly attributed to a variation in the concentration used in the treatment, the initial cell seeding density, different times of treatments and the proliferation potential of the cell line [69].

A family of cysteine proteases (caspases) plays a major role in the implementation of apoptosis in extrinsic and intrinsic pathways [58]. p53 is an important tumour suppressor gene that promotes tumour cell apoptosis and inhibits tumour formation. In addition, p53 is also an important regulator of metabolic homeostasis involved in DNA repair, apoptosis and cell cycle arrest. p53-dependent apoptosis is mainly achieved by influencing mitochondrial function [70]. This study revealed the upregulation of p53 and Bax protein expression in MG and GA treatment as a hallmark of apoptosis-derived mitochondria pathways. Furthermore, both MG and GA triggered the activation of caspase 8 and caspase 9.

Consistent with the potential role of p53 in modulating chemotherapy in human cancers, the loss of p53 function was linked to chemoresistance in certain tumour types. Therefore, the expression of p53 proteins in all treated groups can directly activate the transcription of apoptotic genes to promote apoptosis. The cell cycle arrested at subG0 was correlated to an increase in p53 levels. p53 is involved in a checkpoint that decides whether a cell progresses in the cell cycle or dies by apoptosis in response to deoxyribonucleic acid (DNA) damage [71]. The tumour suppressive capabilities of p53 are related to a variety of stress signals, including DNA damage. Increased expression of the p53 tumour suppressor pathway has been proposed as a novel target pathway for improved cancer therapy [72].

The expression level of apoptotic proteins, such as Bax, caspase 8 and caspase 9, was upregulated. Meanwhile, the Bcl-2, anti-apoptotic protein was correspondingly downregulated, showing consistency with cell cycle events [54]. Induction of apoptosis through caspase 9 activation and involvement of Bax and Bcl-2 lead to the activation of effector caspases, including caspases 3 and 7 [73]. The activation of caspase 8, which is a major participant in the extrinsic pathway, is the result of cell death receptors, such as Fas and TNF-α. Activation of caspase 8 also leads to the activation of effector caspases, including caspases 3 and caspase 7. The effector caspases (3 and 7) were sequentially cleaved and activated by apoptotic stimuli to induce apoptosis in HeLa cells. In addition, the expression of p53 toward HeLa cells suggested that p53 was responsible for the upregulation of Bax and downregulation of Bcl-2 in all treated groups.

The finding in the current study revealed that GA-induced apoptosis by the upregulation of pro-apoptotic Bax and downregulation of Bcl-2 as early as 24 h of treatment [43]. Several previous studies have proven the involvement of the apoptotic mechanism in cell death in cancer cells treated with both MG and GA. Obviously, the majority reported the involvement of intrinsic and extrinsic pathways, such as Bax, were upregulated, whereas executioner caspases, caspase 8 and caspase 9, activity increased. Bcl-2 was downregulated [35,36,39,40,50,51,52,53,55,61,63,65,67].

## 4. Materials and Methods

### 4.1. Gallic Acid and Methyl Gallic

Natural product gallic acid (GA) and methyl gallic (MG) were previously isolated from the ethyl acetate extract of *Quercus infectoria*.

### 4.2. Bioassay-Guided Isolation of Bioactive Compounds from Q. infectoria

Prior to bioassay-guided isolation, finely powdered galls were soaked in an ethyl acetate solution for 72 h. The samples were then filtered, evaporated under reduced pressure and stored at 4 °C until use. Chromatographic separation techniques, thin layer chromatography (TLC) and column chromatography (CC) were utilised to obtain fraction and pure compounds. The first batch of CC yielded four fractions labelled EAQI/FA1, EAQI/FA2, EAQI/FA3 and EAQI/FA4 and tested for an antiproliferative activity to obtain IC_50_.

The fraction with an IC_50_ of less than 50 µg/mL was considered ‘active’. The active fraction (IC_50_ < 50 µg/mL) was selected for further fractionation. Fraction EAQI/FA3 was selected and resumed for a second column chromatography and the result was five subfractions, labeled EAQI/FA3/B1, EAQI/FA3/B2, EAQI/FA3/B3, EAQI/FA3/B4 and EAQI/FA3/B5. Based on the TLC profiles, two pure compounds were detected from fractions EAQI/FA3/B2 and EAQI/FA3/B5. Both compounds were then isolated and eluted with MeOH. The spots on TLC were visualised by a yellow spot in a staining iodine vapour, a dark spot under UV-254 nm and UV-365 nm and an orange spot with ferric chloride (FeCl_3_). Both compounds were further analysed for structure elucidation using Nuclear Magnetic Resonance (NMR) (Bruker, Billerica, MA, USA, Avance at 400 MHz), measuring 1H-NMR and 13C-NMR, COSY, HSQC and HMBC. Chemical shifts were quoted in δ units and the signals were described in terms of the chemical shift, multiplicity, coupling constant (*J*) where applicable and number of protons. The abbreviations *s* (singlet), *d* (doublet), *t* (triplet), *q* (quartet), *m* (multiplet), *dd* (doublet of doublets), *ddd* (doublet of doublets of doublets), *dddd* (doublet of doublets of doublets of doublets), *br s* (broad of singlet) have been used to express multiplicities.

An infrared (IR) (Perkin Elmer, Waltham, MA, USA, FTIR, model 1725X) spectrophotometer using potassium bromide (KBr) discs was used to determine the functional group present in the isolated compound. The vibrational portion of the infrared region was at a wavelength (λ) between 2.5 µm and 25 µm (1 µm = 10^−6^ m). The absorption bands were measured in cm^−1^. The mass of the molecule was determined by Direct Injection Probe (DIP), using Gas Chromatography–Mass Spectrometry (Agilent, Santa Clara, CA, USA, model 7890A), with an ion source temperature of 200 °C and interface temperature of 300 °C.

### 4.3. Cell Culture

HeLa (human cervical cancer cells) and Vero (African Green Monkey kidney cells), a type of normal cell line, were obtained from the American Type Culture Collection (ATCC, Manassas, VA, USA). Both cells were cultured in Dulbecco’s Modified Eagle’s Medium (DMEM) supplemented with 10% foetal bovine serum (FBS) and penicillin–streptomycin 1% (*v*/*v*). The cells were incubated at 37 °C and 5% CO_2_ in a humidified atmosphere.

### 4.4. Antiproliferative Assay

Antiproliferative activity was carried out using 3-(4,5-dimethylthiazol-2-yl)-2,5-diphenyltetrazolium bromide (MTT) tetrazolium reduction [74]. GA, MG and CIS were diluted in DMSO to obtain a stock solution of 10.00 mg/mL. Cells (3 × 10^4^/mL) were treated with various concentrations of compounds in 96-well culture plates for 72 h. After 72 h of exposure, 20 μL of the MTT solution (5 mg/mL in PBS) was added to each well and further incubated for 4 h at 37 °C. Then, the medium was discarded. Insoluble formazan crystals were dissolved by adding 100 µL of DMSO. The plates were shaken, and optical density measured using a microplate reader at 570 nm. Then IC_50_ values of GA and MG were determined by using nonlinear regression analysis (percent inhibition versus concentration) and used in subsequent experiments. In this study, untreated HeLa cells served as negative control and cisplatin (CIS) was used as positive control. All compounds, GA, MG and CIS, were also tested on normal cells, Vero, to confirm the cyto-selective effect toward cancer cells, HeLa.

### 4.5. Cell Morphology Analysis

Apoptotic cell morphology was examined using acridine orange and propidium iodide (AOPI) staining [75]. AOPI are nucleic acid binding dyes that were used for evaluating the changes in nuclear morphology, quantify and qualify the cellular profile of the viable, apoptotic and necrotic cells. AO is a selective fluorescent cationic dye of nucleic acid that generates a protonated positive charge as it crosses the plasma membrane of viable and early apoptotic cells and intercalates to create green fluorescence in DNA and RNA [76]. Viable cells were stained green with an intact nucleus structure. Early apoptotic cells with a bright green-coloured nuclei due to the chromatin condensation. Late apoptotic cells have bright orange areas in the nucleus of condensed chromatin that separate them from necrotic cells. Necrotic cells were stained red by PI, which penetrated the nuclear matter where the cell membrane’s integrity was impaired [77].

HeLa cells (5 × 10^4^ cells/mL) were treated with IC_50_ concentrations of GA, MG and CIS for 24, 48 and 72 h, respectively, in triplicates. The treated cells were trypsinised and harvested with 1 mL cold phosphate buffered saline (PBS), followed by centrifuged at 1500 rpm for 10 min at 4 °C. This process was repeated twice. The cell suspension was mixed with 20 µL of AO/PI solution (1:1) and the mixture (10 mL) was transferred onto a slide and covered with a cover slip. Viable, apoptotic and necrotic cells were quantified in a population of 200 cells using a fluorescence microscope equipped with B-2A filter (Nikon TE2000-U, Tokyo, Japan). Untreated HeLa cells served as negative control and cisplatin was used as positive control.

### 4.6. Determination of Phosphatidylserine (PS) Externalisation

The mode of cell death was investigated further for apoptotic activity by monitoring phosphatidylserine (PS) externalisation, using the annexin V-FITC/PI assay. Early event apoptosis is characterised by the externalisation of PS from the inner layer of the plasma membrane to the outer surface. Quantification of annexin V-FITC binding to externalised PS represents the apoptotic cells. In flow cytometry analysis, the staining of Annexin V/propidium iodide (An/PI) is based on the ability of the Annexin V protein to bind to phosphatidylserine (PS), which is externalised after the induction of apoptosis in the outer cell membrane leaflet. In viable cells, PS is located at the inner membrane leaflet, but on induction of apoptosis, it is translocated to the outer membrane leaflet and becomes available for annexin V binding. The addition of PI enabled viable (An−/PI−), early apoptotic (AnnV+/PI−), late apoptotic (An+/PI+) and necrotic (An−/PI+) cells to be differentiated [78].

The Annexin V-FITC Apoptosis Detection Kit 1 (Beckton Dickinson, Franklin Lakes, NJ, USA) was used to determine phosphatidylserine (PS) externalisation. The kit contains Annexin V conjugated to the fluorochrome FITC, propidium iodide and binding buffer. Briefly, HeLa cells (5 × 10^4^ cells/mL) were treated with an IC_50_ value of the GA, MG and CIS for 3, 6 and 9 h. The cells were collected, washed thrice with cold PBS and 100 µL of binding buffer was added to the tubes. Then, 3 μL of FITC-conjugated Annexin V (Annexin V-FITC) and 3 μL of propidium iodide (PI) was added, and the mixture was then incubated at room temperature (24 °C) in the dark for 15 min. The stained cells were diluted using a binding buffer (400 µL) and immediately analysed with a CytoFlex flow cytometer (Beckman Coulter, Brea, CA, USA). About 10,000 events were accumulated per sample. The results were generated in a quadrant graph with four different populations of cells representing viable cells (Annexin V-FITC and PI were negative), early apoptosis (Annexin V-FITC positive and PI were negative), late apoptosis (Annexin V-FITC and PI were positive) and necrosis (Annexin V-FITC negative and PI were positive). Untreated HeLa cells served as a negative control, and cisplatin was used as a positive control.

### 4.7. Cell Cycle Analysis

Cell cycle analysis was performed according to the protocol in the CycleTESTTM PLUS DNA Reagent Kit (Beckton Dickinson, Franklin Lakes, NJ, USA). HeLa cells (5 × 10^4^ cells/mL) were cultured overnight in a 6-well plate and treated with an IC_50_ value of GA, MG, and CIS for 24, 48 and 72 h at their IC_50_ concentrations in triplicates. The cells were trypsinised and harvested in a similar manner as the AO/PI staining protocol. The cells pellets were resuspended in 250 µL of buffer solution A, followed by incubation for 10 min at room temperature. Then, 200 µL of solution B was added and mixed prior to further incubation for 10 min at room temperature. After that, 200 μL of solution C, containing propidium iodide, was added, and the mixture was incubated in a dark place at 4 °C for 10 min. The cellular DNA content was measured by using a CytoFlex flow cytometer (Beckman Coulter, Brea, CA, USA). Untreated cells were used as a negative control, and cells treated with cisplatin were utilised as a positive control.

### 4.8. Determination of p53, Bax and Bcl-2 Expression

HeLa cells were seeded (5 × 10^4^ cells/mL) in a 6-well plate and incubated for 24 h before being treated with an IC_50_ concentration of MG, GA and CIS. The treated cells were further incubated at 37 °C in a 5% CO_2_ incubator with 90% humidity for 3 h. After that, cells were harvested and washed. The assay was resumed using anti-p53, anti-Bax and anti-Bcl-2 antibody FITC. Staining, fixation and permeabilisation were done following the manufacturer’s protocol.

Briefly, cells were resuspended thoroughly, and 100 μL of the fixative solution was added into each tube and incubated for 15 min at 4 °C. After 15 min, cells were washed twice with 0.2% PBS and Tween 20, followed by adding 100 μL of permeabilisation solution into each tube and incubated for another 15 min at 4 °C. After 15 min, the cells were washed twice with 0.2% PBS and Tween 20. The reactions were blocked using 5% PBS for 15 min and washed again twice with 0.2% PBS and Tween 20. For staining, 10 μL of fluorochrome-conjugated primary antibodies was added to the cells for 20 min at 4 °C in the dark and then washed again twice with 0.2% PBS and Tween 20. The cells were washed twice by centrifugation at 400× *g* for 5 min and resuspended in 500 μL of ice-cold PBS, 10% FCS, and 1% sodium azide. Finally, the cells were analysed using a flow cytometer. Untreated cells were used as a negative control and cells treated with CIS were used as a positive control.

### 4.9. Caspases Analysis

The activities of caspase 8 and caspase 9 were evaluated using the FAM FLICA TM Caspases Kit. The manufacturer’s instructions were followed to perform the analysis. Initially, the FLICA TM was prepared by adding DMSO to solubilise it and diluted with PBS prior to adding it into the sample solution. After 6 h of treatment, disassociated cells from the treated and untreated cell groups of HeLa cells were harvested. The cells were centrifuged at 300× *g* for 10 min and then the supernatant was discarded. After that, 290 μL of fresh media was added. Next, 10 μL of FAM-FLICA working solution was added to the samples at a *v*/*v* ratio of 1:30 and mixed well in the cell suspension to disperse the FAM-FLICA reagent. Then, it was incubated in a CO_2_ incubator at 37 °C for 40 min. After incubation, 2 mL of 1× Apoptosis Wash Buffer was added and gently mixed. Then, the cells were centrifuged at 300× *g* for 10 min. These steps were repeated twice. Then, 300 μL of Apoptosis Wash Buffer was added and placed on ice before proceeding to the flow cytometer analysis. The assay was carried out in three independent replicates for each sample. The data were quantitatively represented as the percentage of activity of caspase 8 and caspase 9 in comparison to the untreated cells group.

### 4.10. Statistical Analysis

All results were obtained from three independent experiments. Data were expressed as the mean ± standard error of the mean (SEM) and analysed by one-way analysis of variance (ANOVA), followed by the Bonferroni post hoc test. The statistical software SPSS (version 22) was used, and *p* < 0.05 was defined as statistical significance compared to the control.

## 5. Conclusions

In conclusion, GA and MG exhibited a potent antiproliferative effect with an apoptosis mode of cell death. Morphological changes and PS externalisation support the induction of an apoptosis event in the treated cells. The percentage of apoptotic cells increased with the time of exposure. However, after 72 h of treatment, the cells became necrotic. Most of the cells at this hour were in the late apoptosis stage.

## Figures and Tables

**Figure 1 ijms-24-08495-f001:**
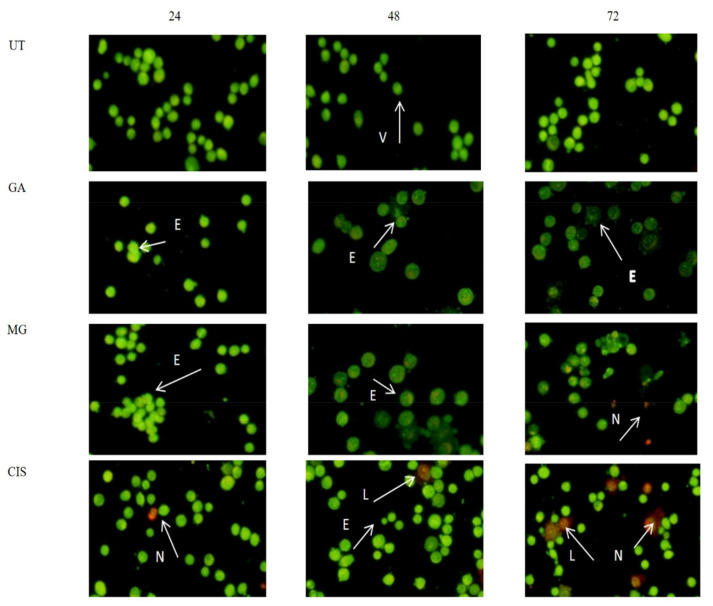
Morphological changes of HeLa cells treated with GA and MG for 24, 48 and 72 h. Untreated cells (UT) were used as a negative control and CIS as a positive control. Arrows indicate viable cell (V), early apoptotic cell (E), late apoptotic cell (L) and necrosis (N). The cells were viewed under a fluorescence microscope (100×).

**Figure 2 ijms-24-08495-f002:**
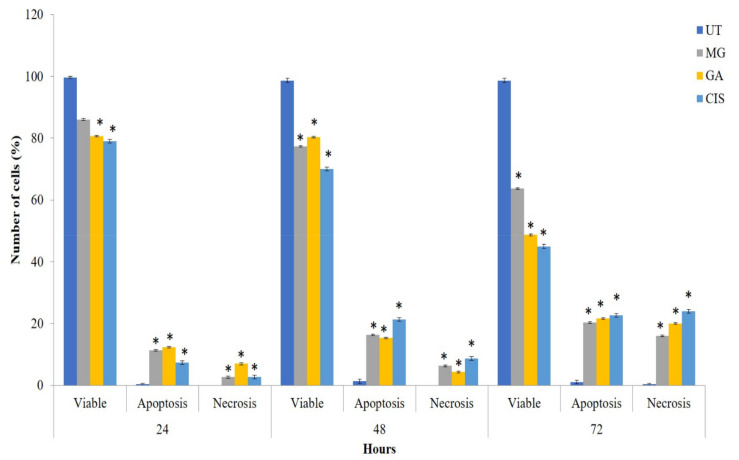
Mode of cell death of HeLa cells after treatment with GA and MG at 24, 48 and 72 h of treatment. CIS was used as a positive control and UT (untreated) group as a negative control. Values are mean ± S.E.M of three independent experiments (n = 3). Asterisk (*) indicates statistically significant (*p* < 0.05) as compared with UT.

**Figure 3 ijms-24-08495-f003:**
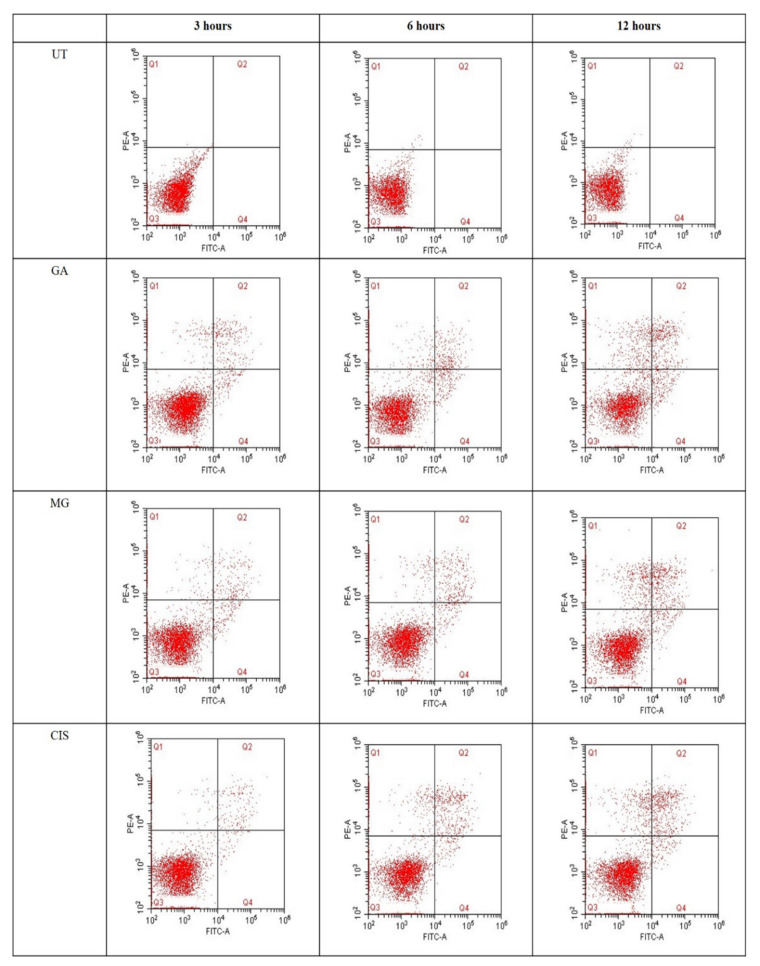
Flow cytometry analysis of HeLa cells after treatment with GA and MG for 3, 6 and 12 h. Scatter plots of FITC-AV/PI double staining in quadrant analysis; Q1: necrosis cells, Q2: late apoptotic cells, Q3: viable cells and Q4: early apoptotic cells. In three independent experiments, CIS was used as control positive and the untreated (UT) group as control negative.

**Figure 4 ijms-24-08495-f004:**
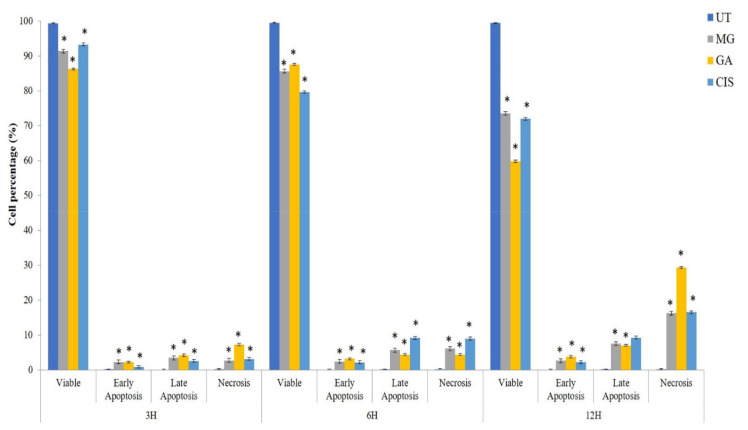
The graph summarises the number of cells in each quadrant after treatment with MG and GA for 3, 6 and 12 h. CIS was used as control positive and the untreated (UT) group as control negative. Values are mean ± S.E.M of three independent experiments (n = 3). Asterisk (*) indicates statistically significant (*p* < 0.05) as compared with the untreated group (UT).

**Figure 5 ijms-24-08495-f005:**
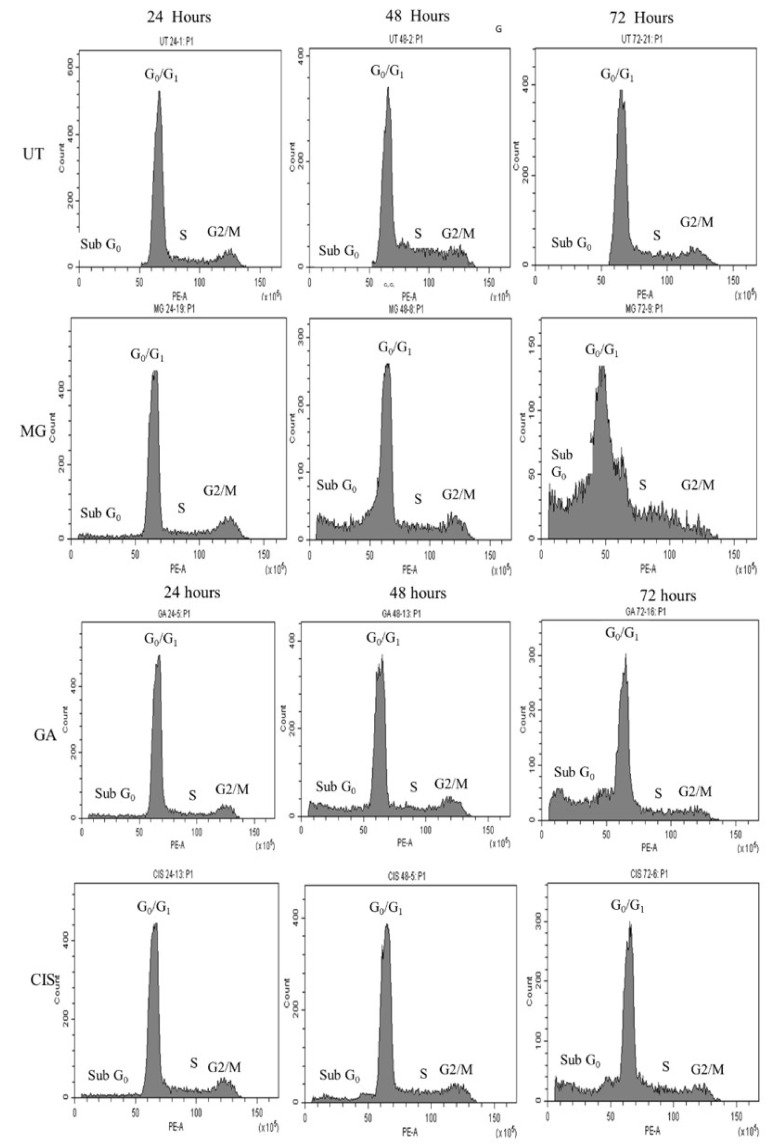
Cell cycle distribution of HeLa cells treated with MG and GA at 24, 48 and 72 h of treatment. In three independent experiments, CIS was used as a positive control and the untreated (UT) group as a negative control.

**Figure 6 ijms-24-08495-f006:**
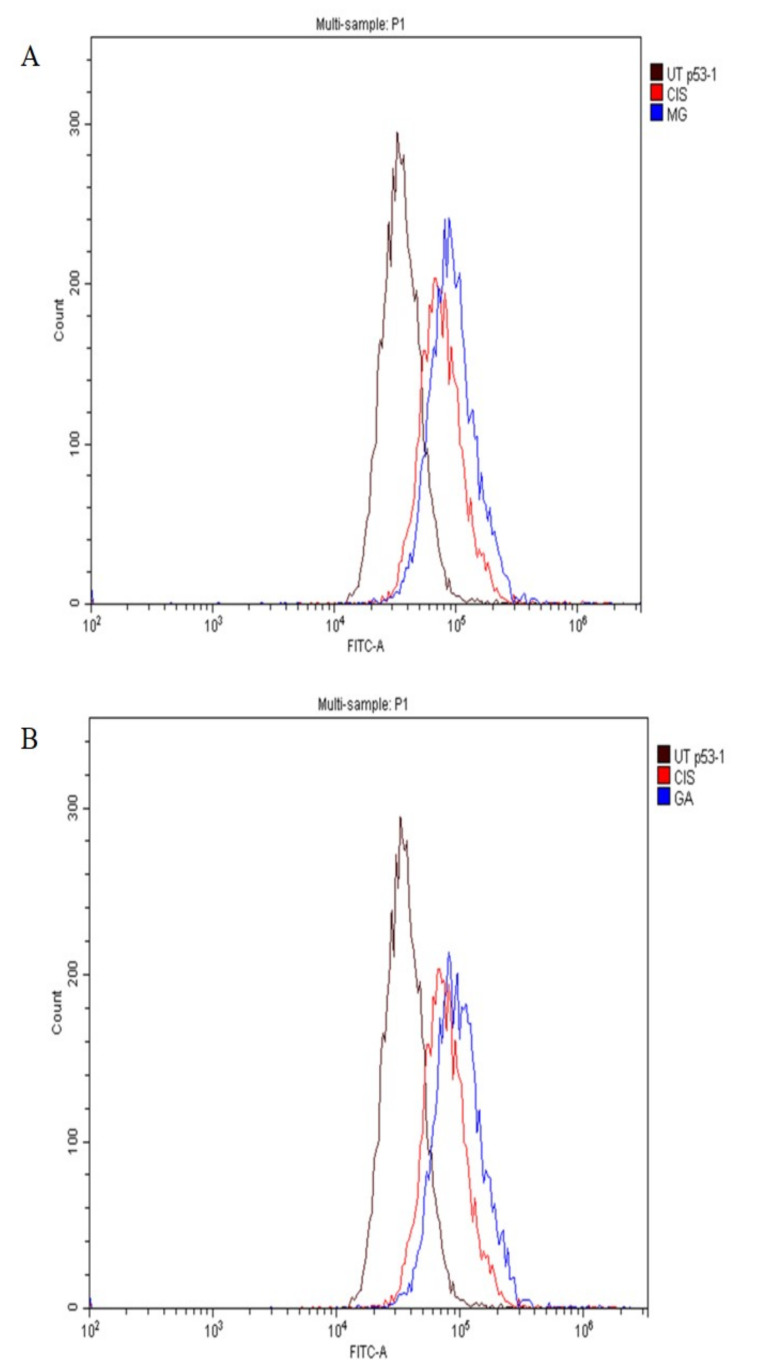
The expression of the p53 protein toward HeLa cells after treatment with MG (**A**), and GA (**B**) for 3 h. CIS was used as a positive control and UT group as a negative control.

**Figure 7 ijms-24-08495-f007:**
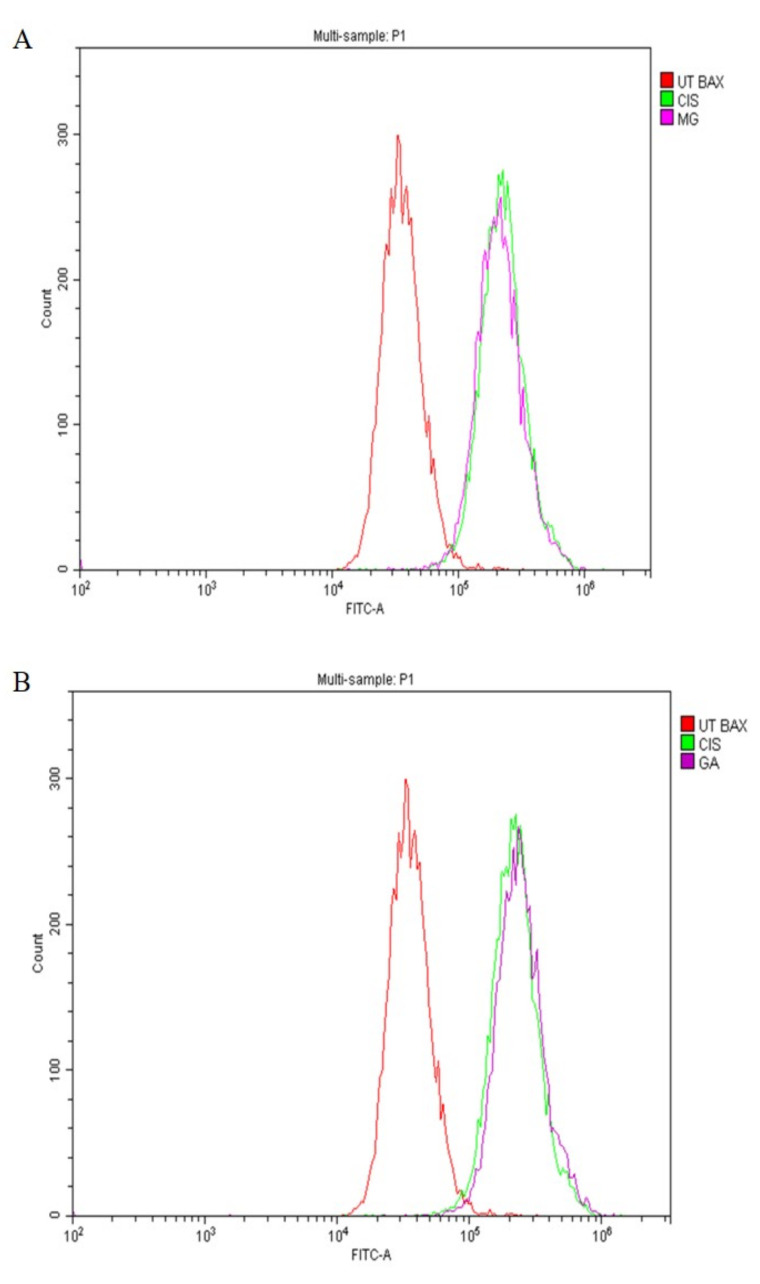
The expression of the Bax protein toward HeLa cells after treatment with MG (**A**), and GA (**B**) for 3 h. CIS was used as positive control and UT group as negative control.

**Figure 8 ijms-24-08495-f008:**
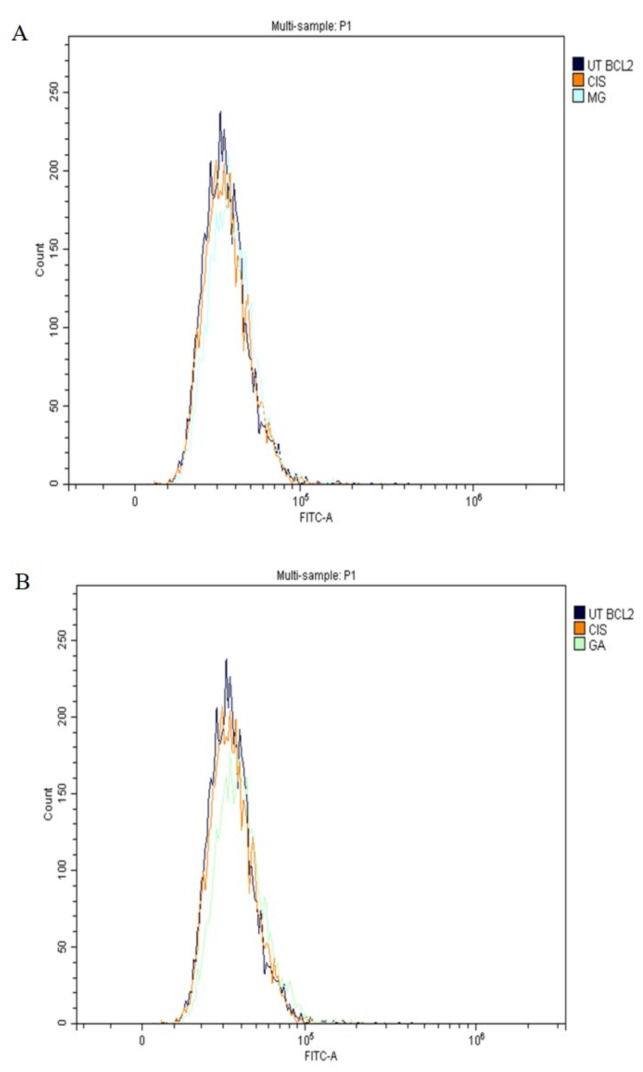
The expression of the Bcl-2 protein toward HeLa cells after treatment with MG (**A**), and GA (**B**) for 3 h. CIS was used as a positive control and UT group as a negative control.

**Figure 9 ijms-24-08495-f009:**
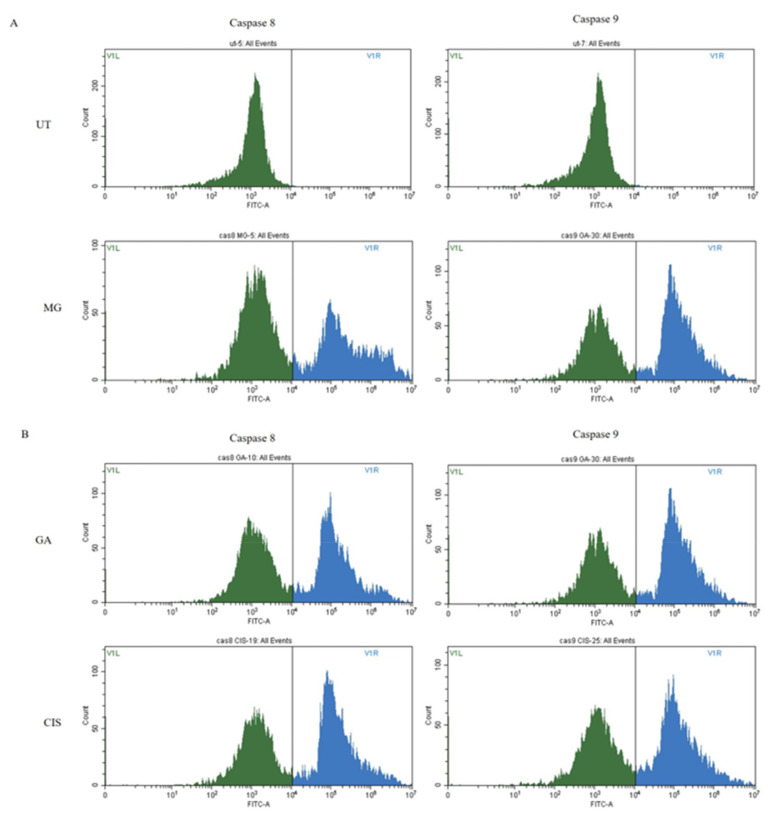
Caspase 8 and caspase 9 activities in HeLa cells treated with MG (**A**), and GA (**B**) for 3 h. Cisplatin was used as a positive control, and UT served as a negative control.

**Table 1 ijms-24-08495-t001:** The IC*_50_* values of the EAQI of HeLa cells. Vero cell lines used as control cells, * *p* < 0.05 was taken as significantly different from positive control (cisplatin).

Compounds	IC_50_ (µg/mL)	
	HeLa	Vero
EAQI	11.50 ± 0.50 *	>100
Cisplatin (CIS)	1.85 ± 0.15	14.33 ± 0.88

**Table 2 ijms-24-08495-t002:** The IC_50_ values of fractions and subfractions from column chromatography on HeLa cells. Vero cell lines used as control cells, * *p* < 0.05 was taken as significantly different from positive control (cisplatin).

Fractions/Subfractions	IC_50_ (µg/mL)		
	HeLa	Vero	
1st CC (Fractions)	EAQI/FA1	>100	>100
	EAQI/FA2	>100	>100
	EAQI/FA3	16.67 ± 1.76 *	>100
	EAQI/FA4	60.33 ± 1.45	>100
	Cisplatin	3.33 ± 0.88 *	13.67 ± 1.20 *
2nd CC (Subfractions)	EAQI/FA3/B1	>100	>100
	EAQI/FA3/B2	11.00 ± 0.58 *	>100
	EAQI/FA3/B3	56.67 ± 1.76	>100
	EAQI/FA3/B4	51.33 ± 0.88	>100
	EAQI/FA3/B5	10.00 ± 1.06 *	>100
	Cisplatin	2.33 ± 0.33 *	11.67 ± 0.88 *

* Active =< 50 µg/mL.

**Table 3 ijms-24-08495-t003:** The IC_50_ concentrations of methyl gallate and gallic acid towards HeLa and Vero cell lines.

Compounds	IC_50_ (µg/mL)	
	HeLa	Vero
Methylgallate (Methyl 3,4,5-trihydroxybenzoate)	11.00 ± 0.58	>100
Gallic acid (3,4,5-trihydroxybenzoic acid)	10.00 ± 1.06	>100
Cisplatin	1.85 ± 0.15	11.67 ± 0.88

## Data Availability

Not applicable.

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
