# Peer review of "Natural Gallic Acid and Methyl Gallate Induces Apoptosis in Hela Cells through Regulation of Intrinsic and Extrinsic Protein Expression"

_ijms, 2023, doi:10.3390/ijms24108495_

Round 1

Reviewer 1 Report

The manuscript describes natural compounds, GA and MG exhibited potent antiproliferative effects with apoptosis mode of cell death. Despite the characterization of the compounds providing some relevant information, I think there is some information that needs to be clarified. Here are my comments:

1.     The introduction part is a bit long and boring. It should be condensed to the core contents according to the keywords of the research that the authors think.

2.     It would be nice if the purpose and methodology of the study could be explained in one paragraph in the last part of the introduction.

3.     Compound information on GA and MG is lacking. Although it was said to have been simply isolated from the EA extract of QI, the research process and result data for this should be presented. The novelty of the compounds GA and MG is lacking, and they are easily commercially available materials. Separation and identification from QI should be added.

Author Response

Thank you for the comments;

  1. The introduction part is a bit long and boring. It should be condensed to the core contents according to the keywords of the research that the authors think. The irrelevant information have been truncated.
  2. It would be nice if the purpose and methodology of the study could be explained in one paragraph in the last part of the introduction. Agree, it has been added.
  3. Compound information on GA and MG is lacking.Although it was said to have been simply isolated from the EAextract of QI, the research process and result data for thisshould be presented. The novelty of the compounds GA andMG is lacking, and they are easily commercially availablematerials. Separation and identification from QI should beadded. The required information has been added

Reviewer 2 Report

This study elucidated the cell death mechanism of gallic acid (GA) and methyl gallate (MG) on human cervical cancer cell line (HeLa) by analyzing the inhibitory activity of gallic acid (GA) and methyl gallate (MG) on 50% cell population (IC50), cell cycle analysis, annexin-V FITC double staining, expression of apoptotic proteins (P53, Bax and Bcl2) and caspase activation. This article has a large workload, fluent language expression and strong innovation. After solving the following problems, we can consider receiving the manuscript.

1. The introduction of gallic acid and methyl gallate in the preface is not sufficient and does not reflect its significance for this study.

2. Please explain how the difference between endogenous pathway and exogenous pathway is reflected.

3. If gallic acid and methyl gallate come from previous separation, please specify the specific source or provide identification certificate.

4. In the text analysis in 2.2, please clearly distinguish the cell morphology analysis of gallic acid or methyl gallate.

5. In Figure 1, the morphology of apoptotic cells shown in the figure is significantly different after the treatment of MG and CIS for 72 hours. Are they all apoptotic cells?

6. The chart definition is poor. Please edit the chart as required.

7. The effect of GA and MG on normal cells is not reflected in the experiment process, but the third paragraph of the discussion has relevant conclusions?

8. In the final part of the results and discussion, special attention should be paid to grammar and sentence structure so that readers can clearly understand the results.

Author Response

  1. The introduction of gallic acid and methyl gallate in the
    preface is not sufficient and does not reflect its significance for
    this study. The introduction of gallic acid and methyl gallate have been added in the section.
  2. Please explain how the difference between endogenous
    pathway and exogenous pathway is reflected. Explanation on endogenous and exogenous pathway has been added.
  3. If gallic acid and methyl gallate come from previous
    separation, please specify the specific source or provide
    identification certificate. The methods for isolation of gallic acid and methyl gallate have been added in the methodology section.
  4. In the text analysis in 2.2, please clearly distinguish the cell
    morphology analysis of gallic acid or methyl gallate. Cells treated with both exhibited same apoptotic features showing the same morphology.

  5. In Figure 1, the morphology of apoptotic cells shown in the
    figure is significantly different after the treatment of MG and
    CIS for 72 hours. Are they all apoptotic cells? All are apoptotic cells.
  6. The chart definition is poor. Please edit the chart as
    required. Chart has been edited.
  7. The effect of GA and MG on normal cells is not reflected in
    the experiment process, but the third paragraph of the
    discussion has relevant conclusions? Yes, it not reflected the experiment but it was to proof the cytoselective effect of the compound towards cancer cells.
  8. The sentences has been rephrased.

Reviewer 3 Report

The manuscript by Abdullah et al. is a well-written and interesting study. There are a few minor comments which will improve the scope of the manuscript.

1)    Authors are suggested to provide a schematic diagram for this article.

2)    Authors are suggested to provide scale in figure 1.

3)    Authors are suggested to increase the resolution of fig. 9.

4) Authors could include western blot technique results to confirm the expression of p53, Bax, and Bcl-2 after treatment with MG & GA.

5) How the authors have determined the concentration of these compounds after extraction and how did they have determined the purity level?

6) Authors are suggested to check the referencing according to the journal format like in 12, 14, 17, etc. full/abbreviated.

7) Please correct the grammatical errors in lines 62 and 332. 

8) Please add the imaging to the methods section, it's missing in the manuscript.

Author Response

  1. The schematic diagram will be added (need more time to prepare, please apologise)
  2. We used microscope electron without scale. Just the magnification was mentioned.
  3. Figure 9 was extracted from the software used together with instrument. The resolution could not be improved.
  4. Thank you for the suggestion. However, the study only used flowcytometry, there was no western blot analysis has been conducted.
  5. IC50 was used as concentration for protein expression. Purity level was determine using spectrometric methods.
  6. The referencing has been improvised according to journal guideline.
  7. Grammatical error has been revised.

Round 2

Reviewer 1 Report

The author's revised paper has been appropriately upgraded based on the reviewer's comments.

This paper can be accepted for publication